# (-)-5-Demethoxygrandisin B a New Lignan from *Virola surinamensis (Rol.) Warb.* Leaves: Evaluation of the Leishmanicidal Activity by In Vitro and In Silico Approaches

**DOI:** 10.3390/pharmaceutics15092292

**Published:** 2023-09-07

**Authors:** Steven Souza Paes, João Victor Silva-Silva, Paulo Wender Portal Gomes, Luely Oliveira da Silva, Ana Paula Lima da Costa, Manoel Leão Lopes Júnior, Daiana de Jesus Hardoim, Carla J. Moragas-Tellis, Noemi Nosomi Taniwaki, Alvaro Luiz Bertho, Fábio Alberto de Molfetta, Fernando Almeida-Souza, Lourivaldo Silva Santos, Kátia da Silva Calabrese

**Affiliations:** 1Institute of Exact and Natural Sciences, Federal University of Pará, Belém 66075-110, PA, Brazil; 2Laboratory of Protozoology, Oswaldo Cruz Institute, Oswaldo Cruz Foundation, Rio de Janeiro 21041-250, RJ, Brazilcalabrese@ioc.fiocruz.br (K.d.S.C.); 3Laboratory of Medicinal and Computational Chemistry, Institute of Physics of São Carlos, University of São Paulo, São Carlos 13418-900, SP, Brazil; 4Collaborative Mass Spectrometry Innovation Center, Skaggs School of Pharmacy and Pharmaceutical Sciences, University of California, San Diego, CA 92123, USA; 5Department of Natural Sciences, Pará State University, Belém 66095-015, PA, Brazil; 6Laboratory of Natural Products for Public Health, Pharmaceutical Technology Institute, Farmanguinhos, Oswaldo Cruz Foundation, Rio de Janeiro 21040-900, RJ, Brazil; 7Electron Microscopy Nucleus, Adolfo Lutz Institute, São Paulo 01246-000, SP, Brazil; 8Laboratory of Immunoparasitology, Oswaldo Cruz Institute, Oswaldo Cruz Foundation, Rio de Janeiro 21040-900, RJ, Brazil; alvaro.bertho@ioc.fiocruz.br; 9Flow Cytometry Core Facility, Oswaldo Cruz Institute, Oswaldo Cruz Foundation, Rio de Janeiro 21040-900, RJ, Brazil; 10Laboratory of Molecular Modeling, Institute of Exact and Natural Sciences, Federal University of Pará, Belém 66075-110, PA, Brazil; fabioam@ufpa.br; 11Postgraduate Program in Animal Science, State University of Maranhão, Sao Luis 65055-310, MA, Brazil

**Keywords:** *Leishmania amazonensis*, mitochondrial membrane potential, flow cytometry, trypanothione reductase, computational studies

## Abstract

Leishmaniasis is a complex disease caused by infection with different *Leishmania* parasites. The number of medications used for its treatment is still limited and the discovery of new drugs is a valuable approach. In this context, here we describe the in vitro leishmanicidal activity and the in silico interaction between trypanothione reductase (TryR) and (-)-5-demethoxygrandisin B from the leaves of *Virola surinamensis* (Rol.) Warb. The compound (-)-5-demethoxygrandisin B was isolated from *V. surinamensis* leaves, a plant found in the Brazilian Amazon, and it was characterized as (7R,8S,7′R,8′S)-3,4,5,3′,4′-pentamethoxy-7,7′-epoxylignan. In vitro antileishmanial activity was examined against *Leishmania amazonensis*, covering both promastigote and intracellular amastigote phases. Cytotoxicity and nitrite production were gauged using BALB/c peritoneal macrophages. Moreover, transmission electron microscopy was applied to probe ultrastructural alterations, and flow cytometry assessed the shifts in the mitochondrial membrane potential. In silico methods such as molecular docking and molecular dynamics assessed the interaction between the most stable configuration of (-)-5-demethoxygrandisin B and TryR from *L. infantum* (PDB ID 2JK6). As a result, the (-)-5-demethoxygrandisin B was active against promastigote (IC_50_ 7.0 µM) and intracellular amastigote (IC_50_ 26.04 µM) forms of *L. amazonensis*, with acceptable selectivity indexes. (-)-5-demethoxygrandisin B caused ultrastructural changes in promastigotes, including mitochondrial swelling, altered kDNA patterns, vacuoles, vesicular structures, autophagosomes, and enlarged flagellar pockets. It reduced the mitochondria membrane potential and formed bonds with important residues in the TryR enzyme. The molecular dynamics simulations showed stability and favorable interaction with TryR. The compound targets *L. amazonensis* mitochondria via TryR enzyme inhibition.

## 1. Introduction

Herbal preparations are plant-based and are often used for the treatment of many diseases. For instance, species from the Myristicaceae family are used by indigenous people of the Amazon region as a source of powerfully hallucinogenic narcotics, triggering chemical and pharmacological interest in the species of this family during the 1950s [1]. Belonging to the Myristicaceae family, *Virola surinamensis* is among the best-known *Virola* species and is popularly known as ucuúba, ucuúba-branca, ucuúba-cheirosa, ucuúba-de-igapó, ucuúba-da-várzea, ucuúba-verdadeira, bicuíba, bicuíba-branca, tree-de-tallow, and virola [2].

Lignans and neolignans have been successfully extracted from the leaves of *V. pavoni*, *V. surinamensis*, *V. michelli*, and *V. sebifera* [3,4,5]. Among these, (-)-5-demethoxygrandisin B arises as a tetrahydrofuran lignan, constituting its first occurrence of isolation within the *Virola* genus. Its structure bears similarity to (-)-5-demethoxygrandisin, another tetrahydrofuran lignan previously isolated from the leaves and tree barks of *V. surinamensis.* The key distinction between (-)-5-demethoxygrandisin and (-)-5-demethoxygrandisin B lies in the configuration of the furan ring. *V. surinamensis* is widely found in Central and South America, including Brazil, and its constituents have been reported with interesting pharmacological properties, such as anti-chagasic, anti-malarial, and leishmanicidal activities [6,7,8,9,10,11]. Additionally, the leaves and seeds of *V. surinamensis* contain a high content of lignans [12], well-known as a promising class of compounds against *Leishmania* spp. [13].

Leishmaniases is a complex of diseases caused by infection with different *Leishmania* parasites from the Trypanosomatidae family. It is transmitted through the bite of infected female sandflies, with diverse clinical manifestations such as a cutaneous, mucocutaneous, and visceral diseases in the Old and New World [14]. Leishmaniasis is considered a neglected disease, and it is estimated that more than 1 billion people are at risk of infection because they live in areas endemic to leishmaniasis [15].

Leishmaniasis predominantly impacts communities already facing vulnerability due to factors such as poverty, limited healthcare access, and various adversities. Finding new medicines represents a humanitarian endeavor aimed at enhancing the well-being of these individuals and guaranteeing their access to treatments that are both impactful and accessible. Moreover, its treatment remains a huge challenge due to serious limitations, such as few effective drugs, resistance, toxicity, high cost, and a lack of adherence to the therapeutic protocol [16]. Thus, leishmaniasis remains a major public health problem worldwide.

Among the molecular targets used to search for new drugs against *Leishmania* and *Trypanosoma*, trypanothione reductase (TryR) has been identified as unique to parasites. Also, TryR has been described as an effective target against trypanosomatids. Thus, that enzyme plays a crucial role in the control of redox homeostasis by the parasite, which is vital for its survival in a hostile environment generated by the host as a response to the infection. The TryR has been an interesting binding site for many drug candidates that exhibit inhibitory activities against parasites [17,18].

Therefore, in order to discover new drugs against *Leishmania*, here we describe (-)-5-demethoxygrandisin B as a potential candidate and its behavior to the TryR enzyme.

## 2. Materials and Methods

### 2.1. Botanical Material, Extraction, and Isolation

The species under study was collected by a parataxonomist from Embrapa Amazônia Oriental (1°26′27.5″ S 48°26′22.8″ W), where an exsiccate from 180,980 was cataloged in the Herbarium. The leaves were dried in an oven at a temperature of 40 °C for a period of two days. After drying, the leaves were crushed and ground in a knife mill, resulting in 6.2 kg of material. The dried and ground leaves of *V. surinamensis* (4.0 Kg) were sequentially extracted by maceration with hexane (Tedia, Fairfield, CT, USA), ethyl acetate (Tedia, Fairfield, CT, USA), and methanol (Tedia, Fairfield, CT, USA) for a period of seven days for each solvent. The obtained masses of the hexane, EtOAc, and methanol extracts were 61.7 g, 120 g, and 244 g, respectively. EtOAc was selected to exhibit a superior chromatographic profile in thin-layer chromatography. A 6.0 g portion of the EtOAc extract was dissolved in 1 L of MeOH/H_2_O solution (7:3), followed by successive partitioning with 700 mL of hexane (×4) and 700mL of ethyl acetate (×4). After drying with anhydrous sodium sulfate, both yielded 0.2 g and 2.8 g, respectively. The ethyl acetate phase (2.8 g) was submitted to a fractionation on column chromatography using Silica gel 60 (70–230 mesh) and 230–400 mesh (SILICYCLE, Québec, QC, Canada) as the stationary phase and polarity gradients of acetone in hexane and methanol in acetone, as elution systems. The subfractions were analyzed by thin-layer chromatography (TLC) (silica gel F254, Merck, Darmstadt, Germany) using Hexane/EtOAc (6:4) as the elution system, and ceric sulfate was used as a detection spray. A yellowish oil, obtained in 30% hexane/acetone, was isolated and characterized as the (-)-5-demethoxygrandisin B: (7R,8S,7′R,8′S)-3,4,5,3′,4′-pentamethoxy-7,7′-epoxylignan (40 mg). Obtaining the extracts and the substance (-)-5-demethoxygrandisin B is presented in Figure 1.

### 2.2. Spectroscopy of HRMS and NMR for Structural Characterization, and Optic Rotation Determination

The assessment of (-)-5-demethoxygrandisin B at 100 ppm in methanol: H_2_O (80:20) was conducted using a Xevo^®^ G2-S QTof coupled with an ACQUITY Ultra Performance LC™ system (Waters Corp., Milford, MA, USA). The ionization source was configured with a desolvation gas flow (N2) at 600 L/h and a desolvation temperature of 150 °C. The cone gas flow (N2) was set at 50 L/h, and the source temperature was maintained at 120 °C. Adjustments were made to the capillary and sampling cone voltages, setting them at 1.0 kV and 40 V, respectively. The data acquisition was executed utilizing MassLynx 4.1 software (Waters, Milford, USA) [19,20]. NMR 1D (^1^H and ^13^C) and 2D homonuclear-correlated spectroscopy (COSY ^1^Hx^1^H), HMBC, and HSQC analyses were acquired with a Bruker 400 spectrometer AscendTM (Rheinstetten, Germany) model at 400.15 MHz (^1^H) and 100.62 MHz (^13^C). The chemical shifts were determined relative to CDCl_3_ at 0 ppm. A total of 20 mg of (-)-5-demethoxygrandisin B was solubilized in 600 μL of CDCl_3_. TopSpin 3.6.0 software was used for data control and processing. Thereby, the spectra were manually evaluated. The optical rotation of the (-)-5-demethoxygrandisin was measured on a Perkin–Elmer 341 polarimeter (Perkin–Elmer Inc., Waltham, MA, USA).

### 2.3. Ethical Statements and Animals

The implementation of animal procedures adhered to the guidelines established by the National Council for Control of Animal Experimentation (Conselho Nacional de Controle de Experimentação Animal—CONCEA) and received endorsement from the institutional Ethics Committee on Animal Care and Utilization (Comissão de Ética no Uso de Animais do Instituto Oswaldo Cruz—CEUA-IOC L53/2016-A3). Female BALB/c mice aged 4 to 6 weeks were purchased from the Institute of Science and Technology in Biomodels of the Oswaldo Cruz Foundation.

### 2.4. Peritoneal Macrophage Isolation and Parasite Cultures

Thioglycollate-elicited peritoneal macrophages were obtained from BALB/c mice that were injected intraperitoneally 72 h previously with 3 mL of sterile thioglycollate broth as described by Silva-Silva et al. [21]. Then, cells were maintained in RPMI 1640 medium and cultured overnight at 37 °C, under a humidified atmosphere of 5% CO_2_. Promastigotes of *L. amazonensis* H21 (MHOM/BR/76/MA-76) were cultivated at 26 °C in Schneider’s Insect medium (Sigma, St Louis, MO, USA). All media were supplemented with 10% fetal bovine serum (Gibco, Gaithersburg, MD, USA), 100 IU/mL penicillin, and 100 μg/mL streptomycin as previously described [22].

### 2.5. Cytotoxicity Assay

To determine the cytotoxic effects of (-)-5-demethoxygrandisin B, the peritoneal macrophages were cultured in 96-well plates (5 × 10^5^ cells/mL) with different concentrations of (-)-5-demethoxygrandisin B (31–994 μM) at least in triplicate up to a final volume of 100 μL per well in 5% CO_2_ for 24 h at 37 °C. Wells without cells were used as blanks, and wells with cells and dimethyl sulfoxide (DMSO) 1% were used as controls. All assays were performed in triplicate at three different times using amphotericin B (0.20–27 μM) as the reference drug. Cell viability was measured using a colorimetric methyl thiazole tetrazolium (MTT) assay [23]. The results were used to calculate the 50% cell cytotoxicity (CC_50_).

### 2.6. Antileishmanial Activity Assay and Selectivity Index

The promastigote forms of *L. amazonensis* were used at a 10^6^ parasites/mL concentration from a 3- to 5-day-old culture. The assay was performed in 96-well plates in the presence of different concentrations of (-)-5-demethoxygrandisin B at 16–497 μM, in a final volume of 100 µL per well, and was then incubated for 24 h at 26 °C. Wells with medium and without parasites were used as blanks, and wells with parasites incubated with DMSO 1% were used as controls. Amphotericin B at 0.0169–1 μM was used as the reference drug. The parasite viability post-treatment was ascertained by enumerating live promastigotes, factoring in flagellar motility, employing Neubauer’s camera in conjunction with an optical light microscope [24]. This count was compared with the score of the nontreated promastigote growth. The results were expressed as a parasite growth inhibitory concentration of 50% (IC_50_). The effect of (-)-5-demethoxygrandisin B on the intracellular amastigote form was performed in 24-well plates, with coverslips, with peritoneal macrophages (5 × 10^5^ cells per well) cultured and infected with promastigote forms of *L. amazonensis* using a ratio of 10:1 parasite per cell. After 6 h, the cells were washed three times with PBS to remove the free parasites. The infected cells were treated with (-)-5-demethoxygrandisin B at 8–124 μM, or amphotericin B at concentrations spanning 0.169 to 2.7 μM, over a 24 h period. Following this, the coverslips containing the treated and infected cells were fixed using Bouin solution and subsequently stained with Giemsa for subsequent light microscopy analysis. The determination of the IC50 values ensued from quantifying the intracellular amastigotes within 100 host cells, adopting the methodology outlined by Silva-Silva et al. [22]. These experiments were carried out in independent triplicate with each condition performed in triplicate. The selectivity index (SI) was obtained from the ratio of the peritoneal macrophages CC_50_ and IC_50_.

### 2.7. Nitrite Quantification

The impact of (-)-5-demethoxygrandisin B on nitric oxide (NO) release was assessed indirectly through the quantification of nitrite in the culture supernatants of macrophages (5 × 10^6^ cells/mL) using the Griess reaction. Subsequent to treatment with (-)-5-demethoxygrandisin B (62 μM) and/or stimulation with *L. amazonensis* (3 × 10^7^ parasites/mL) or LPS (10 μg/mL), approximately 50 μL of the supernatants was collected and introduced into designated wells of 96-well plates. To these supernatants, 50 µL of Griess reagent (comprising 25 µL of 1% sulfanilamide in a 2.5% H_3_PO_4_ solution and 25 µL of 0.1% N-(1-naphthyl) ethylenediamine solution) was added. Following a 10 min incubation, the plates were subjected to spectrophotometric analysis at 570 nm, and the nitrite levels were deduced through reference to the standard sodium nitrite curve (ranging from 1.5 to 100 μM) [21].

### 2.8. Transmission Electron Microscopy

*L. amazonensis* promastigotes treated with IC_50_ for (-)-5-demethoxygrandisin B and incubated for 24 h at 26 °C were collected by centrifugation at 5000 rpm for 5 min. The parasites underwent fixation for an overnight duration within a solution comprising 2.5% glutaraldehyde (Sigma, St Louis, MO, USA) immersed in 0.1 M sodium cacodylate buffer (pH 7.2). After this, a sequence of three washes utilizing 0.1 M sodium cacodylate buffer followed. Subsequently, a post-fixation procedure was carried out through immersion in a solution containing 1% osmium tetroxide, 0.8% potassium ferrocyanide, and 5 mM calcium chloride. Following further washing steps in 0.1 M sodium cacodylate buffer, the parasites underwent dehydration through a graduated acetone series before being embedded within EPON 812 resin (Sigma, St Louis, MO, USA). Ultrathin sections, spanning 100 nm, were then attained via Sorvall MT 2-B (Porter Blum) ultramicrotome (Sorvall, Newtown, CT, USA). These sections were subsequently stained with a 5% uranyl acetate aqueous solution and lead citrate (comprising 1.33% lead nitrate and 1.76% sodium citrate) prior to observation under a transmission electron microscope JEM-1011 (JEOL, Tokyo, Japan) operating at 80 kV [25].

### 2.9. Determination of Mitochondrial Membrane Potential (MMP) (ΔΨm)

For the determination of the mitochondrial membrane potential in the promastigote forms of *L. amazonensis*, we used tetramethylrhodamine ethyl ester (TMRE) (Molecular Probes, Carlsbad, CA, USA) and flow cytometry. For this method, 2 × 10^6^ parasites/mL were treated with (-)-5-demethoxygrandisin B IC_50_ for 24 h at 26 °C. Nontreated parasites were used as a negative control, and heat-killed parasites (60 °C bath for 30 min) were used as a positive control. Post-incubation, the parasites were subjected to centrifugation at 1500× *g* for 5 min at room temperature. Afterward, they were washed with PBS and incubated in a solution of 300 μL TMRE (50 nM) in the absence of light for 15 min at room temperature. Subsequently, a flow cytometry analysis was conducted using a CytoFlex flow cytometer (Beckman Coulter Life Sciences, Inc., Brea, CA, USA). TMRE excitation was achieved using a 488 nm blue laser, and the emitted fluorescence was captured with a 585/42 nm bandpass filter. The flow cytometry data were analyzed using CytExpert software version 2.1 (Beckman Coulter Life Sciences, Inc., Brea, CA, USA) [26].

### 2.10. Molecular Docking

The enzyme trypanothione reductase (TryR) is a promising target for the design of new drugs with antileishmanial activity, because it plays a fundamental role in the intracellular redox balance necessary for the survival of parasites of the *Leishmania* genus and is not present in the host system, where it is replaced by its human homologous enzyme (glutathione reductase) [17]. In this sense, the docking calculations were performed using the GOLD program (Genetic Optimization for Ligand Docking) version 5.5 [27] with the crystallographic structure of the TryR enzyme from *L. infantum*, retrieved from the Protein Data Bank (PDB ID 2JK6) with a resolution of 2.95 Å and complexed by the FAD cofactor [28]. GOLD is a docking program that uses the genetic algorithm to explore the conformational flexibility of the ligand in a complex with a partially flexible macromolecule [29]. As the TryR enzyme is a homodimer [30] composed of two identical monomers, the A chain present in the enzyme structure was selected for the docking step, while the B chain was removed in the Chimera program [31]. To perform the docking, the two-dimensional structure of neolignan (-)-5-demethoxygrandisin B was previously constructed in the MarvinSketch program (https://chemaxon.com/presentation/marvin-the-nextgeneration-of-chemical-drawing, accessed on 10 July 2020) and later optimized using the molecular mechanic’s force field MMFF94 [32] in the Avogadro program [33] for the in silico study of the compound. Subsequently, the GOLD program removed all the water molecules, and hydrogen atoms were added to the TryR enzyme. Then, the binding site was defined based on the location of the residues that participate in the catalytic mechanism (Cys52, Cys57, His461′, and Glu466′) [34]. Finally, the composite was docked by applying the GoldScore and ChemScore scoring functions with a search efficiency of 100% in GOLD. The GoldScore and ChemScore functions provide rapid results that predict binding affinity with the macromolecule [35]. Because interactions with the amino acid residues of the active site are essential to promote the inhibition of enzyme activity, a visual inspection of the interactions of the more thermodynamically stable conformation of (-)-5-demethoxygrandisin B was performed on the online server PoseView [36,37].

### 2.11. Molecular Dynamics

Initially, the calculation of the electrostatic potential charge for the structure obtained during docking was performed by the Gaussian 03 program [38], applying the restricted electrostatic potential (RESP) [39] combined with the Hartree–Fock method together with the basis (6–31 G(d,p)) [40]. The protonation states of all the amino acid residues present in the enzyme were determined under pH 7.0 by the PROPKA server [41]. The molecular dynamics (MD) simulations were performed by the AMBER 18 program, implemented by the *pmemd.CUDA* module [42]. The general AMBER force field (GAFF) [43] was used to describe the (-)-5-demethoxygrandisin B ligand, while the MMFF99SB force field was used for the amino acid residues of the enzyme [44]. Using the *tleap* module of the AmberTools18 package, the enzyme–ligand complex was solvated with the TIP3P explicit solvent model in a cubic box with an edge equal to 12 Å, followed by the addition of Cl- counterions to neutralize the resulting charges [45]. The system went through 4 energy minimization steps by the SANDER module [46], applying 10,000 steps in the first step with the steepest descent method and 15,000 with the conjugate gradient method, with the following 3 steps being carried out in 10,000 steps evenly split for both methods. Subsequently, the system was gradually heated from 0K to 298K in the NVT set, using the Langevin algorithm [47] with the restriction of the atoms by a force constant equal to 25 kcal/mol·Å^2^. The periodic boundary conditions employed by the Particle Mesh Ewald (PME) method [48] were used to identify long-range electrostatic interactions. The limit condition of 1 atm and the temperature of 298 K were used to maintain constant temperature and pressure set NPT. The SHAKE algorithm [49] was employed to constrain all hydrogen bonds. The equations of motion were integrated every 2 femtoseconds (2 fs) using the Verlet algorithm [50] based on the principle of finite differences. Thus, the enzyme–ligand complex was simulated at a time of 50 ns by the NPT set at a temperature of 298 K. In addition, the CPPTRAJ module [51] available in AMBERTOOLS 18 was applied for a stability analysis, in terms of the root mean square deviation (RMSD) calculation and structural flexibility through the B-factor.

#### Binding Free-Energy Calculations

The molecular mechanics Poisson–Boltzmann surface area (MMPBSA) [52] and molecular mechanics generalized Born surface area (MMGBSA) [53] methods were used to calculate the binding free energy (ΔG_bind_) from the complex to the last 10 ns of the MD trajectory, extracted using the CPPTRAJ and MMPBSA.py modules available in AmberTools 18 [54]. Thus, the binding free energy was calculated according to the following equations:ΔG_bind_ = ΔE_MM_ + ΔG_bind,solv_ − T.ΔS(1)
ΔE_MM_ = ΔE_int_ + ΔE_ele_ + ΔE_vdW_
(2)
ΔEint = ΔE_bond_ + ΔE_angle_ + ΔE_torsion_(3)
ΔG_bind,solv_ = ΔG_GB/PB_ + ΔG_n-polar_
(4)
ΔG_n-polar_ = γ × SASA (5)
ΔG_bind_ = ΔG_vdW_ + ΔG_ele_ + ΔG_GB_ + ΔG_n-polar_
(6)

The term ΔE_MM_ in Equation (1) represents the molecular mechanical energy; ΔG_bind_, solve is the free energy of solvation; and the term T.ΔS represents the product of the absolute temperature and conformational entropy. In Equation (2), the individual contributions to the mechanical energy are presented, composed of the internal (ΔE_int_), electrostatic (ΔE_ele_), and van der Waals (ΔE_vdW_) energy. The free energy of solvation (ΔGb_ind,solv_) results from the sum of polar (ΔG_PB/GB_) and non-polar (ΔG_n-polar_) free energy, as shown in Equation (3). The polar electrostatic contribution was calculated by the Poisson–Boltzmann (PB) and by the generalized Born approximation (GB) methods, while the non-polar energy was estimated by the product between the empirical parameters of the surface tension (γ) and solvent accessible area (SASA), presented in Equation (4). Finally, the interaction free energy obtained in the last 10 ns of the simulation by the MMGBSA method was decomposed according to Equation (6), for the analysis of the individual contribution of the amino acid residues. This method has been widely applied in drug design studies due to its satisfactory approach that successfully reproduces experimental results. The main amino acid residues that contribute to the total energy MMGBSA were visualized using the CHEWD plugin, to generate a color-coding model that expresses the magnitude of the contribution by residues [55].

### 2.12. Statistical Analysis

The numerical results were expressed as the mean ± standard deviation, and the statistical analyses were conducted through the statistical software GraphPad Prism^®^ version 7 (GraphPad Software Inc., San Diego, CA, USA). Differences were considered significant when *p* < 0.05.

## 3. Results

### 3.1. Lignan Structure Elucidation

The (-)-5-demethoxygrandisin B (40 mg) was isolated from the ethyl acetate extract and showed yellowish oil characteristics. High-resolution mass spectrometry (HRMS) data provided a molecular formula C_23_H_30_O_6_ with a calculated and measured precursor monoisotopic mass of 403.2126 Da, corresponding to an error of 0.00 ppm (Appendix A). After that, it was confirmed by spectroscopic experiments of ^1^H and ^13^C nuclear magnetic resonance (NMR) spectra. The ^1^H, ^13^C, heteronuclear single-quantum correlation (HSQC), and heteronuclear multiple-bond coherence (HMBC) NMR data are available in Table 1 and in Appendix A.

The (-)-5-demethoxygrandisin B showed oxybenzyl proton absorptions H-7 and H-7’ (*δ* 4.65, *d*, *J* = 4.5 Hz, ^1^H and 4.63, *d*, *J* = 4.5 Hz, ^1^H) and methyl protons H-9 (*δ* 1.09, *d*, *J* = 6 Hz, 3H) and H-9’ (*δ* 1.06 (*d*, *J* = 6 Hz, 3H) related to tetrahydrofuran. The presence of oxybenzyl proton was confirmed by an analysis of the literature on the classes of substances isolated from *V. surinamensis* [11,56]. Based on the ^1^H NMR spectrum, (-)-5-demethoxygrandisin B showed common signals to tetrahydrofuran lignan. This hypothesis was confirmed once signals were observed in the region of aromatic hydrogens at *δ_H_* 6.96 (*d*, *J* = 1.8 Hz, ^1^H), 6.92 (*dd*, *J* = 1.8 and 8.1 Hz, ^1^H), and 6.85 (*d*, *J* = 8.1 Hz, ^1^H) in an AMX coupling system, assigned to the hydrogens of a 1,3,4-trisubstituted aromatic ring, with the COSY correlation map showing the existing correlations between these signals (Appendix A).

An analysis of the aromatic hydrogen region of the ^1^H NMR spectrum of the (-)-5-demethoxygrandisin B leads to the conclusion that the two aromatic rings were 3′,4′-dimethoxyphenyl and 3,4,5-trimethoxyphenyl. Interpretations of the ^1^H and ^13^C NMR, HSQC, and HMBC spectra for (-)-5-demethoxygrandisin B are described in Table 1, where the carbon signals at *δ_C_* 88.35 and 88.47, as well as those at *δ_C_* 51.06 and 50.97, were typical, respectively, of carbons C-7, C-7’, C-8, and C-8’ of tetrahydrofuran lignans [12,57]. Thus, it was possible to infer that the profile of the ^1^H NMR spectrum of (-)-5-demethoxygrandisin B is common to tetrahydrofuran lignan, which has two phenylpropane units linked by an oxygen atom as illustrated in Figure 2. The stereochemistry of the furan ring was identified by analyzing the coupling constants (*J*) with data well-established in the literature presented in the discussion of this paper [9,28,56,58,59,60].

### 3.2. Antileishmanial Activity and Cytotoxicity

In vitro antileishmanial assays were performed to evaluate the effect of the (-)-5-demethoxygrandisin B in the promastigote and intracellular amastigote forms of *L. amazonensis*, as well as its cytotoxic effect against mammal cells. The (-)-5-demethoxygrandisin B exhibited inhibitory activity against both forms in a concentration-dependent manner (Figure 3). (-)-5-demethoxygrandisin B was able to inhibit the promastigote forms of *L. amazonensis* with a 50% inhibitory concentration (IC_50_) of 7.0 µM. Furthermore, (-)-5-demethoxygrandisin B was active against the intracellular amastigote forms, displaying IC_50_ values of 26.04 µM. These results suggest that promastigotes were more sensitive to (-)-5-demethoxygrandisin B treatment. The cytotoxic effect on peritoneal macrophages showed a CC_50_ value of 193.37 µM, lower than the values observed for the standard drug amphotericin B. The selectivity index (SI) for promastigotes and intracellular amastigotes showed that (-)-5-demethoxygrandisin B was 26.6 and 7.4 times less toxic for macrophages than parasites, respectively. The reference drug amphotericin B showed antileishmanial activity and cytotoxicity as expected, after 24h of treatment (Table 2).

### 3.3. Nitrite Quantification in L. amazonensis-Infected Peritoneal Macrophages Treated with (-)-5-Demethoxygrandisin B

As shown in Figure 4, the NO levels were indirectly estimated by nitrite quantification in the supernatant of BALB/c peritoneal macrophages. An increase in the nitrite levels in the supernatant of cells treated with (-)-5-demethoxygrandisin B (5.75 ± 0.24 μM NaNO_2_, *p* = 0.1587) when compared to untreated cells (5.21 ± 0.21 μM NaNO_2_) was observed, although this increase was not statistically significant. This effect remained even when macrophages were infected with *L. amazonensis* and treated with (-)-5-demethoxygrandisin B (6.67 ± 1.53 μM NaNO_2_, *p* = 0.8286) when compared to stimulated and untreated cells (5.69 ± 0.20 μM NaNO_2_). Macrophages stimulated with lipopolysaccharide (LPS) showed the expected higher levels of nitrite compared to cultures not stimulated with LPS.

### 3.4. Ultrastructural Changes

A transmission electron microscopy analysis revealed untreated parasites displaying organelles with no alterations (Figure 5A), while the main alteration induced by (-)-5-demethoxygrandisin B treatment was kinetoplast swelling (Figure 5B), with alteration in a kDNA compacting pattern (yellow arrow). In addition, vacuoles (asterisks) near the flagellar pocket, vesicular electron-dense structures with an outer bilayer membrane (thin arrows), a structure that looks like an autophagosome (thick arrow), and enlargement of the flagellar pocket with electron-dense filaments inside were also observed in the (-)-5-demethoxygrandisin B-treated parasite.

### 3.5. Mitochondrial Membrane Potential (Δψm) by Flow Cytometry

To investigate the influence of treatment with (-)-5-demethoxygrandisin in the mitochondrial membrane potential of *L. amazonensis* by flow cytometry, 2 × 10^6^ promastigotes/mL were incubated with 50 nM tetramethylrhodamine ethyl ester (TMRE), a cell-permeant fluorescent dye that is readily sequestered by active mitochondria. For this, in Figure 6, we represent the flow cytometric protocol used to assess the frequency of *Leishmania* promastigotes with mitochondrial activity (TMRE+), where a dot plot of morphological patterns of size and granularity (forward scatter (FSC) versus side scatter (SSC)) was created to define the population of *L. amazonensis* promastigotes (Region P4) (Figure 6A). From this region, we created a TMRE-PE-fluorescence histogram to quantify the TMRE+ promastigotes; Figure 5B,C represent an unstained promastigotes sample and (-)-5-demethoxygrandisin B-untreated-*L. amazonensis* promastigotes, which showed 83.52 ± 4.55% of the TMRE+ promastigotes, respectively. The inhibition of the mitochondrial membrane potential of the heat-treated (60 °C) parasites showed a significant decrease in the Δψm to 49.42 ± 1.60% (*p* = 0.0286) (Figure 6D). Statistically significant changes in the mitochondrial membrane potential were also observed after treatment with 7.0 µM of (-)-5-demethoxygrandisin B (38.76 ± 0.67%, *p* = 0.0286) (Figure 6E,F).

### 3.6. Molecular Docking

To assess whether (-)-5-demethoxygrandisin B has potential antileishmanial activity, the enzyme trypanothione reductase (TryR) from *Leishmania infantum* was obtained from the Protein Data Bank (PDB) database under code 2JK6 [28] and selected as a molecular target for the docking simulation. The GoldScore and ChemScore scoring functions were used to predict the binding affinity of the more stable configuration of (-)-5-demethoxygrandisin B in the redox catalytic site (Cys52, Cys57, His461′, and Glu466′) and generated the respective scores of 52.94 and 40.76. The predicted binding modes on the docking for (-)-5-demethoxygrandisin B in the *L. infantum* TryR enzyme cavity were analyzed in the PoseView online server. As shown in Figure 7, it can be seen that the conformation of the molecule interacts with the enzyme through hydrophobic interactions with the Thr51, Cys57, and Leu334 amino acids residues and hydrogen bonds with the Lys60 and Asp327 residues.

### 3.7. Molecular Dynamics

From the coordinates obtained in the molecular docking procedures, the more stable configuration of (-)-5-demethoxygrandisin B was submitted to 50 ns of molecular dynamics (MD) simulations. The root mean square deviation (RMSD) values were calculated to evaluate the structural stability of the enzyme–ligand complex. The plot of the RMSD (Å) versus simulation time (ns) (Figure 8) demonstrates that the TryR-ligand complex stabilized after 20 ns, with a mean RMSD value of 2.63 Å and standard deviation ± 0.41. Thus, it is possible to infer that the compound remained in the enzyme cavity, showing few conformational changes in the complex’s structure.

To evaluate the flexibility of the protein regions concerning (-)-5-demethoxygrandisin B, the B-factor analysis was carried out. From Figure 9, we can note the complex generally presented low values of flexibility, except the region formed to the Ser76-Asn91 amino acid sequence (highlighted in red in Figure 9), which is the most flexible region of the enzyme.

### 3.8. Binding Free-Energy Calculations

The results in Table 3 show that the binding free energy by the molecular mechanics generalized Born surface area (MMGBSA) method for the TryR-(-)-5-demethoxygrandisin B complex was −30.41 ± 2.79 kcal/mol, while the ΔG_bind_ value by the molecular mechanics Poisson–Boltzmann surface area (MMPBSA) was calculated to be -27.66 ± 3.55 kcal/mol. Furthermore, by comparing the binding free-energy values of these two methods, it is possible to observe that both are favorable and stable for the formation of the complex. Note that the van der Waals energy (ΔE_vdW_) presents the greatest contribution to the total free energy, indicating that it acts as a determining factor for the affinity of (-)-5-demethoxygrandisin B for the active site of the TryR enzyme. Furthermore, electrostatic energy (ΔE_ele_) and non-polar solvation energy (ΔE_n-polar_, GB/ΔE_n-polar, PB_) also showed favorable contributions, while polar solvation energy (ΔG_solv, GB_/ΔG_solv, PB_ ΔE_PB_) and the solvation free-energy total (ΔG_solv, GB_/ΔG_solv, PB_) showed unfavorable contributions.

### 3.9. Per-Residue Energy Decomposition 

Energy decomposition by residue was employed to identify the TryR enzyme residues involved in the interactions with (-)-5-demethoxygrandisin B, based on the energy contribution calculated by the MMGBSA method for the last 10 ns of the MD simulations. The binding free-energy values below -1 kcal/mol were defined as a criterion for the selection of amino acid residues that performed the most favorable interactions, that is, that contributed to the stabilization of the ligand in the complex. An analysis of the graph (Figure 10) indicates that Cys52 (−1.17 kcal/mol), Gly326 (−1.18 kcal/mol), Asp327 (−1.86 kcal/mol), Leu334 (−2.20 kcal/mol), and Thr335 (−1.59 kcal/mol) contributed most significantly to the total free energy of the complex.

## 4. Discussion

(-)-5-demethoxygrandisin B was detected as a protonated molecule [M + H]^+^ of m/z 403.2126 and error = 1.24 ppm. These mass spectrometry data presented high precision and accuracy in the high-resolution mass. Additionally, ^1^H and ^13^C NMR spectra to (-)-5-demethoxygrandisin B showed signals attributed to a tetrahydrofuran lignan. Thus, the molecular formula to (-)-5-demethoxygrandisin B was correctly attributed as C_23_H_30_O_6_. The stereochemistry was established based on the literature data [56], i.e., the methyl groups have a *cis* relationship with the aromatic rings, because when methyl groups present a *cis* relation, the shift expected is approximately or greater than at *δ_H_* 1.0, close to the signals revealed to (-)-5-demethoxygrandisin B (*δ_H_* 1.06 and 1.09).

Given these considerations, it is possible to conclude that the signals in *δ_H_* 4.67–4.63 are two doublets, one in *δ_H_* 4.65 (*d*, *J* = 4.5 Hz, ^1^H) for H-7 and the other in *δ_H_* 4.63 (*d*, *J* = 4.5 Hz, ^1^H) for H-7’. The coupling constant (*J* = 4.5 Hz) is by the Karplus vicinal correlation [58], where these hydrogens are expected to have a smaller dihedral angle with their respective vicinal hydrogens H-8 and H-8’, in the gauche position. Lopes et al. [12], when isolating 3,4,5,3’,4’-pentamethoxy-5-demethoxygrandisin, assigned to carbons C-7 and C-7’ the signals at *δ_C_* 88.36 and 88.49, respectively. Therefore, the carbon signals at *δ_C_* 88.35 and 88.47 of the ^13^C NMR spectrum of (-)-5-demethoxygrandisin B were attributed to carbons C-7 and C-7’, respectively. To determine the conformation of the tetrahydrofuran ring of (-)-5-demethoxygrandisin B, the four chiral centers of the substance (C-7, C-7’, C-8, C-8’) were considered, so there are several possible isomers; however, when performing projections and taking into account the coupling constant of H-7 and H-7’ (*J* = 4.5 Hz), where vicinal hydrogens are expected to be in the *cis* position, only two possible pairs of enantiomers remain.

^13^C NMR studies carried out by Fonseca [57] concluded that tetrahydrofuran lignans with the ferryl group in the equatorial position present a C-1 or C-1’ carbon signal shift at *δ_C_* 134.6, while for (-)-5-demethoxygrandisin B it is observed that the C-1’ carbon showed a signal at *δ_C_* 134.86, suggesting that the aryl group attached to C-1’ is in the pseudoequatorial position. Fonseca [57] also carried out ^13^C NMR studies for the methyl groups of the aforementioned lignans, where the methyl groups presented the following displacements for carbon: in structure I at *δ_C_* 13.7; in structure II at *δ_C_* 12.9; and in structure III at *δ_C_* 11.6. Therefore, it is observed that when the methyl groups are in the *trans* position there is a greater chemical shift for the respective carbon signals. For (-)-5-demethoxygrandisin B, it was observed in the ^13^C NMR spectrum that the methyl groups presented signals at *δ_C_* 13.83 and 14.07, thus concluding that they are in the *trans* position. Thus, there are the following possibilities for determining the absolute configuration of (-)-5-demethoxygrandisin B: 7S,8R,7’S,8’R or 7R,8S,7’R,8’S. As the substance presented an optical rotation [α]D = −11.15° (MeOH) and according to the studies byKubanek et al. [59], Kubanek et al. [60], Hwang et al. [61], and Biftu et al. [62], tetrahydrofuran lignans with absolute configuration 7S,8R,7’S,8’R present optical rotation (+), while with absolute configuration 7R,8S,7’R,8’S present optical rotation (−); therefore, for (-)-5-demethoxygrandisin B which showed optical rotation (−), the absolute configuration of said substance is 7R,8S,7’R,8’S. This is where the meeting of the cited analyses allowed us to conclude that (-)-5-demethoxygrandisin B is the lignan (7R,8S,7′R,8′S)-3,4,5,3′,4′-pentamethoxy-7,7′-epoxylignan (Figure 2).

Excitedly, the tetrahydrofuran lignan (-)-5-demethoxygrandisin B isolated in *V. surinamensis* leaves demonstrated leishmanicidal activity. Grandisin, another tetrahydrofuran neolignan found in *V. surinamensis* and *Piper solmsianum*, exhibits potent trypanocidal activity against trypomastigote forms of *Trypanosoma cruzi* and shows promising potential for applications in medicine as an antileishmanial agent [63]. The literature data report that lignans have excellent potential as growth inhibitors for the promastigote and intracellular amastigote stage of *Leishmania* [64,65]. The present data showed that the selectivity index for (-)-5-demethoxygrandisin B was more prominent for *Leishmania* than for macrophages. Ten lignans isolated from the hexane-ethyl acetate extract of *Phyllanthus amarus* leaves showed that the CC_50_ for all the samples was >100 µg/mL, thus revealing low cytotoxicity against macrophages and selectivity against the parasite [66].

(-)-5-demethoxygrandisin B showed greater selectivity against the promastigote stage of *L. amazonensis* compared to the intracellular amastigote, demonstrating the direct action of this compound on this parasite. This can also be attributed to the lower compound bioavailability within the macrophage, which may have resulted in reduced activity against the intracellular amastigote. An indirect mechanism involved with the inhibition of intracellular amastigotes is related to macrophage activation, especially the induction of nitric oxide (NO) [22]. However, there were no significant changes in this assessment parameter when we analyzed the nitrite quantification of *L. amazonensis*-stimulated peritoneal macrophages treated with (-)-5-demethoxygrandisin B.

Therefore, to understand the leishmanicidal activity of the (-)-5-demethoxygrandisin B against promastigote forms, observations of ultrastructural and morphological alterations were performed as a direct way to determine parasite inhibition. Our observations revealed the presence of numerous large vacuoles, vesicular structures with electron-dense content and outer bilayer membranes, autophagosome-like structures, and an enlargement of the flagellar pocket with electron-dense filaments inside it. However, the most significant alteration was observed in the mitochondrion/kinetoplast, which displayed swelling and an altered kDNA compacting pattern. *Leishmania* parasites, like other species of trypanosomatids, have a single mitochondrion with specific energetic and antioxidant enzymes that regulate oxidative stress and bioenergetics and a unique arrangement of mitochondrial DNA (kinetoplast DNA). The kDNA is visible as a prominent disk-shaped spot located in the paraflagellar region of the single elongated mitochondrion, and its gene encoding information is far from that set needed for mitochondrial processes, revealing its importance in the regulation of pro- and antioxidant processes not only in the mitochondria [67,68]. The mitochondrial differences between mammals and trypanosomatids make this organelle a natural drug target, and the observed ultrastructural alteration led us to evaluate the mitochondrial membrane potential of parasites treated with (-)-5-demethoxygrandisin B.

Our results show that (-)-5-demethoxygrandisin B induced the depolarization of the mitochondrial membrane of the promastigote parasite, suggesting inhibition of the metabolic activity of parasites. *L. amazonensis* promastigotes treated with compounds like lignans change the mitochondrial membrane potential and lower ATP production [66]. Impairment of the membrane potential allows compounds to cross the mitochondrial membrane which may lead to a loss of the impermeability to intracellular electrolytes and consequently leads to inhibition of ATP production, resulting in parasite death [69,70,71]. This decrease in mitochondrial membrane potential can occur due to the oxidative imbalance caused by competitively inhibiting TryR activity, a pathway responsible for the leishmanicidal effect [72].

The TryR enzyme plays a crucial role in the antioxidant defense of trypanosomatids and has been widely used in molecular modeling studies in the literature [73,74]. Tetrahydrofuran lignans isolated from *V. surinamensis* showed potent trypanocidal activity against the trypomastigote form of *T. cruzi* [11]. A study by Oliveira et al. [75] suggests that the inhibition of *T. cruzi* TryR is related to the trypanocidal action of these lignans. Thus, to try to understand the inhibition mechanism of the compound against *Leishmania*, the enzyme TryR from *L. infantum* (PDB ID 2JK6) was selected as a molecular target for the docking simulation.

In this sense, the links with the residues Thr51, Lys60, Asp327, and Leu334 observed in the docking are considered important for biological activity against the TryR enzyme, as they participate in the binding of the FAD cofactor at its respective site. It is worth noting that the catalytic mechanism of the TryR occurs due to the transfer of electrons from NADPH via FAD to the two catalytic cysteines (Cys52 and Cys57), and the possible interference in this process prevents the reduction in the disulfide bridge [34]. Furthermore, the interaction with Cys57 is crucial for the potential inhibition of the enzyme as it makes the inactivation of the catalytic cysteine possible and thus stops the nucleophilic attack on Cys52 necessary for the release of the reduced substrate (T[SH]2) [17]. Therefore, it is observed that the more stable configuration obtained by the molecular docking of (-)-5-demethoxygrandisin B has an affinity for the amino acid residues of the active site and performs interactions with a catalytic cysteine considered important for the enzymatic activity.

In evaluating the MD simulations, it was found that the complex exhibited a high B-factor value due to its location in the N-terminal portion of an α-helix loop [76]. The studies carried out by Kuldeep et al. [74] report that the structure of the TryR enzyme of *L. infantum* shows characteristic fluctuations in the N- and C-terminal regions, mainly caused by the presence of unstable loops in these positions.

The binding affinity between the studied compound and the macromolecule was determined by calculating the binding free energy, which is an important tool for designing potent enzymatic inhibitors [77]. The amino acids Cys52, Gly326, Asp327, Leu334, and Thr335 contributed to the stabilization of the (-)-5-demethoxygrandisin B ligand in the complex. These key residues played an important role in stabilizing the ligand in the active site of the TryR enzyme, where they are located. Furthermore, the MD results corroborate those obtained in the molecular docking, which also identified interactions with Gly326, Asp327, and Leu334. Residues Gly326, Asp327, and Leu334 are part of the FAD cofactor binding site, whose interactions are also considered relevant for molecular recognition and the binding affinity of inhibitors [78]. The contribution of Cys52 suggests potential bioactivity of the more stable configuration of (-)-5-demethoxygrandisin B against the TryR enzyme, as this binding prevents the reaction of cysteine with the T[S]2 substrates for the production of mixed disulfide and, consequently, inhibits enzymatic activity [79]. A potent inhibitor studied by Baiocco et al. [28] presents a tetrahedral geometry at the redox site, whose active conformation is oriented by binding to catalytic residues and Thr335, demonstrating the significant role of this amino acid for the antileishmanial activity of drug candidates. A study carried out by Feitosa et al. [80] showed from the interaction energy analysis per residue that the contributions of Gly326 and Thr335 are responsible for the permanence of the ligand in the TryR site. Therefore, the results corroborate previous studies investigating the trypanothione catalytic mechanism and provide a theoretical basis for the design of lignan (-)-5-demethoxygrandisin B as a potential inhibitor of the TryR enzyme of *L. infantum*.

## 5. Conclusions

The (7R,8S,7′R,8′S)-3,4,5,3′,4′-pentamethoxy-7,7′-epoxylignan, namely (-)-5-demethoxygrandisin B, isolated from the leaves of *V. surinamensis* showed to be selective for the parasites, inhibiting the promastigote and intracellular amastigote forms of *L. amazonensis*. The antiparasitic effect of this compound was confirmed by the ultrastructural changes with decreased mitochondrial membrane potential in *L. amazonensis* promastigotes. The docking analyses and molecular dynamics simulation showed a possible mechanism of inhibition promoted by a more thermodynamically stable conformation of (-)-5-demethoxygrandisin B to the TryR enzyme.

## Figures and Tables

**Figure 1 pharmaceutics-15-02292-f001:**
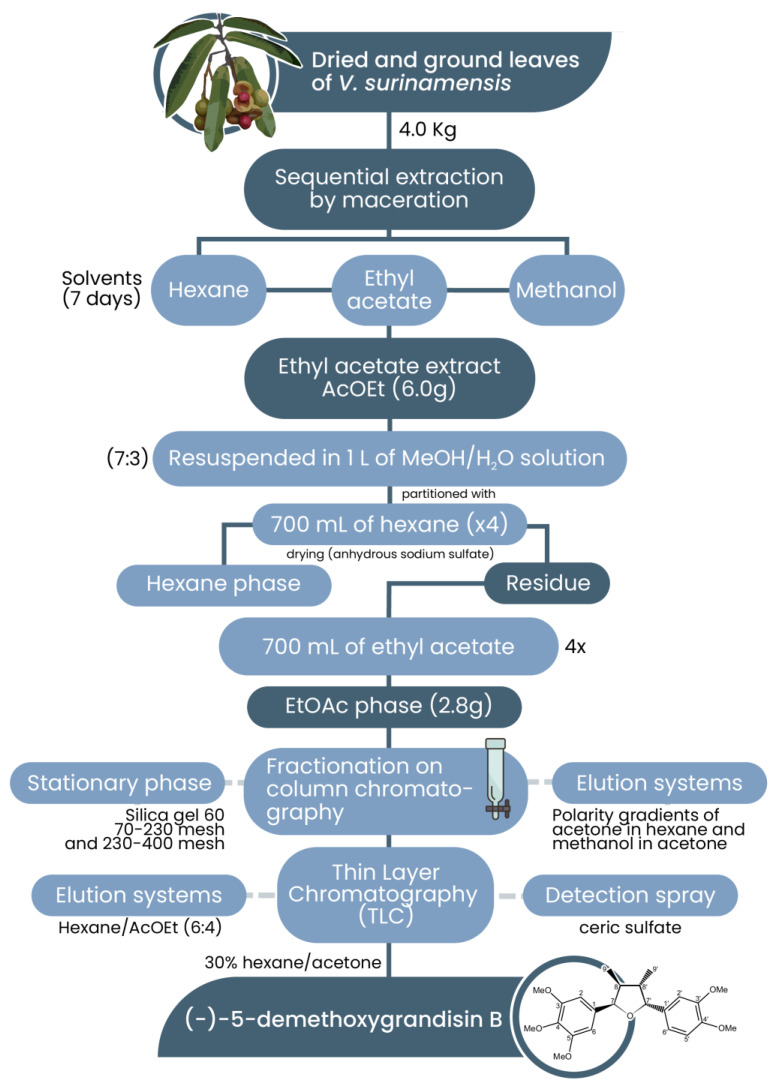
Flowchart of obtaining the extracts and (-)-5-demethoxygrandisin B.

**Figure 2 pharmaceutics-15-02292-f002:**
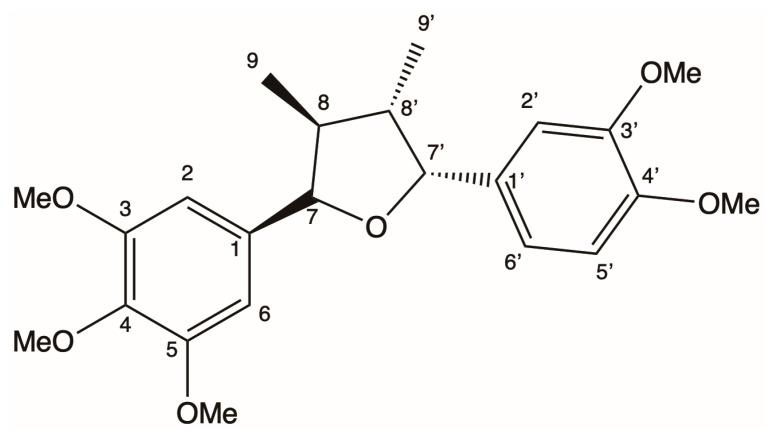
Chemical structure of the (-)-5-demethoxygrandisin B.

**Figure 3 pharmaceutics-15-02292-f003:**
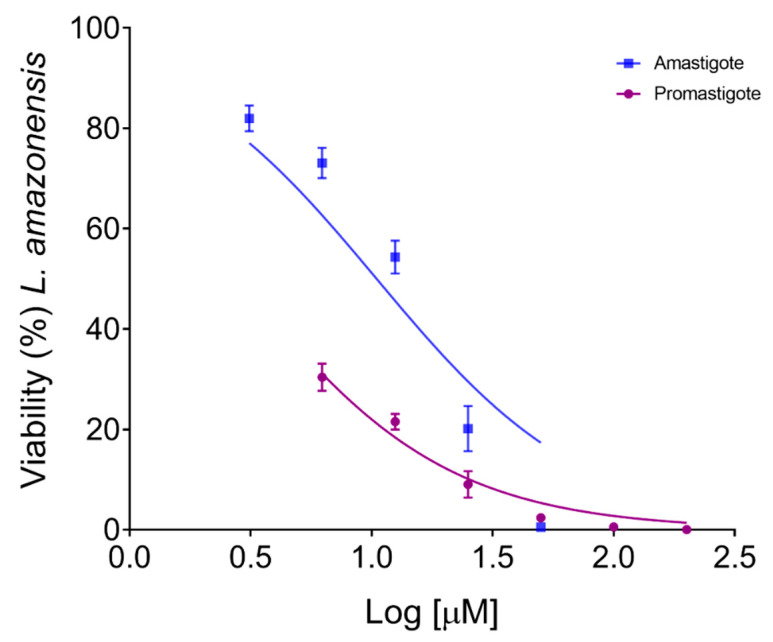
Concentration–response curve of (-)-5-demethoxygrandisin B effects on the viability of *Leishmania amazonensis* promastigote and intracellular amastigote forms. Data represent the mean ± standard error of two independent experiments carried out in triplicate.

**Figure 4 pharmaceutics-15-02292-f004:**
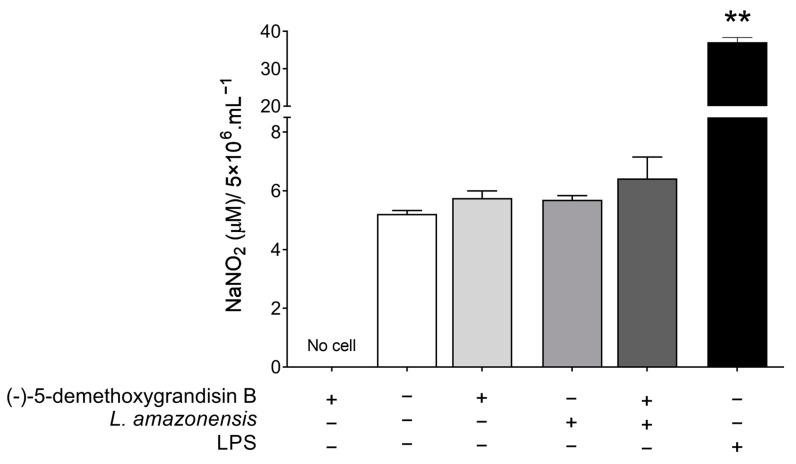
Nitrite quantification in the supernatant of BALB/c peritoneal macrophages treated with compound (-)-5-demethoxygrandisin B at 62 μM, stimulated or not with *Leishmania amazonensis*. LPS (lipopolysaccharide from *Escherichia coli*; 10 μg/mL) was used as the positive control. ** *p* < 0.01 when compared with the untreated group by Mann–Whitney test.

**Figure 5 pharmaceutics-15-02292-f005:**
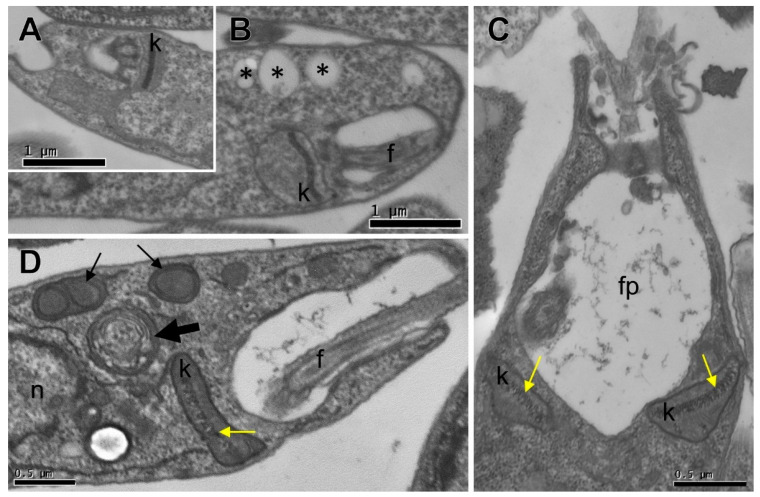
Transmission electron microscopy analysis of *Leishmania amazonensis* promastigote forms treated with (-)-5-demethoxygrandisin B at 25 ug/mL for 24 h. (**A**) Untreated parasites display organelles with no alterations. (**B**–**D**) Parasites treated with (-)-5-demethoxygrandisin B compound exhibit kinetoplast swelling (**B**), with altered kDNA compacting pattern (yellow arrow), vacuoles (asterisks) near flagellar pocket, vesicular electron-dense structures with outer bilayer membrane (thin arrows), a structure that looks like an autophagosome (thick arrow), and enlargement of flagellar pocket. k: kinetoplast; f: flagellum; fp: flagellar pocket; n: nucleus.

**Figure 6 pharmaceutics-15-02292-f006:**
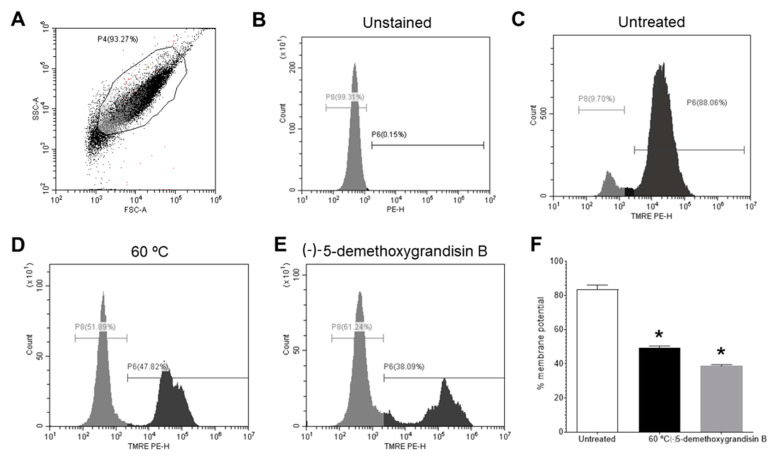
Flow cytometry of mitochondrial membrane potential (ΔΨm) in promastigote forms of *Leishmania amazonensis* treated with (-)-5-demethoxygrandisin B. (**A**) FSC versus SSC dot plot to define *L. amazonensis*-promastigotes population. (**B**) TMRE-staining histogram of control samples (unstained and untreated parasites) gated on “promastigotes.” (**C**) TMRE-staining histogram of (-)-5-demethoxygrandisin B-untreated parasites. (**D**) *L. amazonensis* promastigotes treated by heat (60 °C). (**E**) Representative histogram of *L. amazonensis* promastigotes treated with IC_50_ of (-)-5-demethoxygrandisin B. (**F**) Statistically significant differences were observed between the percentages of cells marked with TMRE in the untreated group and the groups treated with (-)-5-demethoxygrandisin B, at the IC_50_ concentration (7.0 µM). * *p* < 0.05, when compared with the untreated group by Mann–Whitney test. Images are representative of two independent experiments carried out at least in triplicate.

**Figure 7 pharmaceutics-15-02292-f007:**
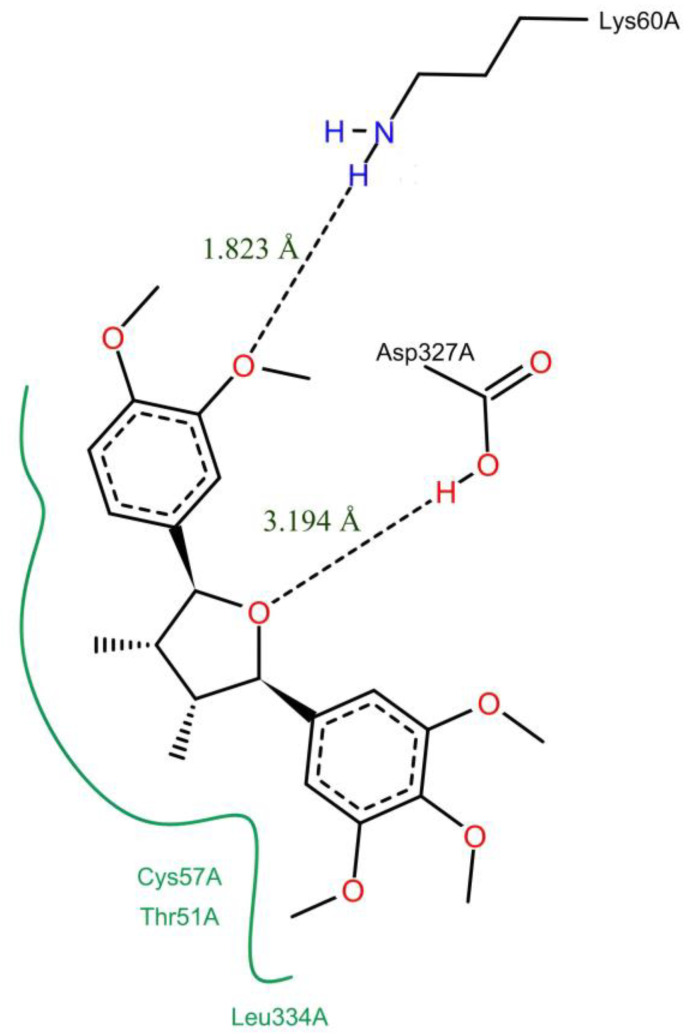
Interactions predicted from docking simulations between the more stable configuration of (-)-5-demethoxygrandisin B and TryR enzyme. Hydrogen bond interactions are represented by dashed black lines, and hydrophobic contacts are represented by continuous green lines. The figure was generated by the PoseView online server.

**Figure 8 pharmaceutics-15-02292-f008:**
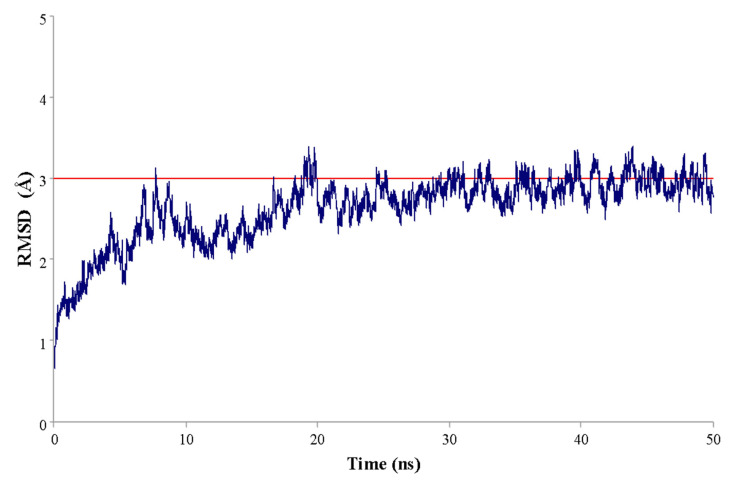
Graphical representation of the root mean square deviation (RMSD) values versus simulation time for the studied complex.

**Figure 9 pharmaceutics-15-02292-f009:**
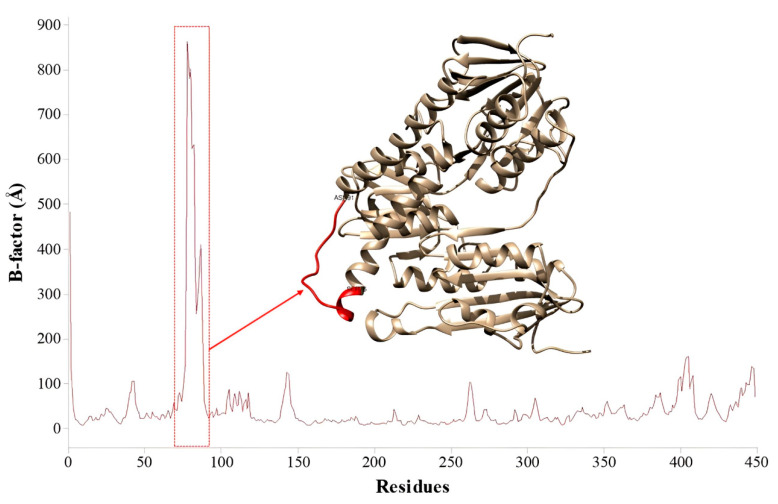
Graphical representation of the structural fluctuations taking into account the enzyme’s residues from the molecular dynamics simulations.

**Figure 10 pharmaceutics-15-02292-f010:**
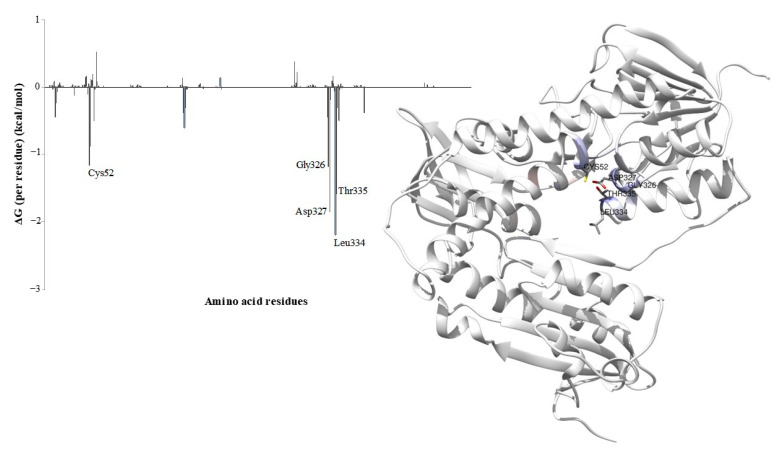
Interaction plot by residues obtained by the MMBGSA method. On the right, the residues that most contributed to the binding free energy, visualized by the CHEWD plugin in the Chimera program.

**Table 1 pharmaceutics-15-02292-t001:** ^1^H, ^13^C, HSQC, and HMBC NMR data from the (-) 5-demethoxygrandisin B.

No.	δ*^1^_H_* in ppm (CDCl_3_) (400 MHz)	*δ_C_* in ppm (100 MHz)	HMBC
1	-	137.46	-
2	6.63 (*s*)	103.14	88.47 (3*J*)
3	-	153.27	-
4	-	138.12	-
5	-	153.27	-
6	6.63 (*s*)	103.14	153.27 (2*J*) 138.12 (3*J*) 137.87 (3*J*)
7	4.65 (*d*, 4.5 Hz)	88.35	-
8	1.80–1.78 (*m*)	51.06	88.35 (2*J*)
9	1.09 (*d*, 6.0 Hz)	14.07/13.83	-
1′	-	134.86	-
2′	6.96 (*d*, 1.8 Hz)	109.27	148.55 (2*J*)
3′	-	148.60	-
4′	-	149.14	-
5′	6.85 (*d*, 8.1 Hz)	110.97	134.86 (3*J*)
6′	6.92 *dd* (1.8 and 8.1 Hz)	118.61	109.27 (3*J*)
7′	4.63 (*d*, 4.5 Hz)	88.47	-
8′	1.80–1.78 (*m*)	50.97	88.35 (2*J*)
9′	1.06 (*d*, 6.0 Hz)	14.07/13.83	88.47 (3*J*)
OMe-3/5	3.88 (*s*)	56.18	153.27 (2*J*)
OMe-4	3.83 (*s*)	60.83	138.12 (2*J*)
OMe-3′	3.87 (*s*)	55.95	148.60 (2*J*)
OMe-4′	3.91 (*s*)	55.95	149.14 (2*J*)

Multiplicities and coupling constants (*J*) in Hertz are shown in parentheses. HSQC: Heteronuclear Single-Quantum Correlation; HMBC: Heteronuclear Multiple-Bond Coherence; NMR: Nuclear Magnetic Resonance.

**Table 2 pharmaceutics-15-02292-t002:** IC_50_ and CC_50_ values (µM) for (-)-5-demethoxygrandisin B and its selectivity index (SI) against promastigote and intramacrophage amastigote forms of *Leishmania amazonensis*.

Compounds	Cytotoxicity	*L. amazonensis*			
	CC_50_ (µM)	Promastigote		Intracellular Amastigote	
		IC_50_ (µM)	SI_pro_	IC_50_ (µM)	SI_ama_
(-)-5-demethoxygrandisin B	193.37	7.0	26.6	26.04	7.4
Amphotericin B	8.82 µM	0.02226	396.1	0.1898	46.5

Data represent mean ± SD of at least two experiments realized in triplicate; CC_50_: cytotoxic concentration for 50% of peritoneal macrophages; IC_50_: inhibitory concentration for 50% of parasites; SI: selectivity index about cytotoxicity for the BALB/c peritoneal macrophages.

**Table 3 pharmaceutics-15-02292-t003:** Binding free energy (ΔG_bind_) of the TryR-(-)-5-demethoxygrandisin B more stable configuration complex and the respective energy components obtained with the MMGBSA/PBSA methods.

MMGBSA		MMPBSA	
Contribution	TryR-(-)-5-Demethoxygrandisin B (kcal/mol)	Contribution	TryR-(-)-5-Demethoxygrandisin B (kcal/mol)
ΔE_vdw_	−45.57 (2.33)	ΔE_vdw_	−45.57 (2.33)
ΔE_ele_	−5.10 (2.70)	ΔE_ele_	−5.10 (2.70)
ΔE_GB_	26.00 (3.17)	ΔE_PB_	29.95 (3.87)
ΔE_n-polar, GB_	−5.74 (0.25)	ΔE_n-polar, PB_	−6.94 (0.24)
ΔG_solv, GB_	20.26 (3.08)	ΔG_solv, PB_	23.01 (3.83)
ΔG_bind, GB_	−30.41 (2.79)	ΔG_bind, PB_	−27.66 (3.55)

The standard deviations are given in parentheses. MMPBSA: molecular mechanics Poisson–Boltzmann surface area; MMGBSA: molecular mechanics generalized Born surface area; ΔE_vdW_: van der Waals energy; ΔE_ele_: electrostatic energy; ΔE_GB_: polar solvation energy by the generalized Born method; ΔE_n-polar, GB/PB_: non-polar solvation energy; ΔG_solv, GB/PB_: total solvation free energy; ΔG_bind, GB/PB_: total free energy.

## Data Availability

All supporting data used in this study are available from the authors.

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
