# Peer review of "(-)-5-Demethoxygrandisin B a New Lignan from Virola surinamensis (Rol.) Warb. Leaves: Evaluation of the Leishmanicidal Activity by In Vitro and In Silico Approaches"

_pharmaceutics, 2023, doi:10.3390/pharmaceutics15092292_

Round 1
Reviewer 1 Report (Previous Reviewer 1)
Thanks for answering my comments. Now it looks good to me and can be published as is.
Author Response
Thank you for taking the time to review our manuscript and for providing feedback. We are glad to hear that the revised version meets your expectations and is suitable for publication as it is.
Reviewer 2 Report (New Reviewer)
The authors report the isolation, characterization and biological evaluation as leishmanicide of a new lignan isolated from the leaves of the plant Virola surinamensis (Rol.) Warb native to Central and South America.
The manuscript has been well structured, easy to understand, the reported data have been clearly described and discussed.
Based on the above, my opinion is that the manuscript should be accepted, but first the authors should take into account the following recommendations.
1. Indicate in Table 1, what is the meaning of the letter "a" (400 MHz)a, as well as the letter J in the values reported for the HMBC experiment.
2. If the authors are sure that the secondary metabolite obtained has a negative optical rotation, they should first place the instrument in which it was performed and second place a sign (-) before the name of the characterized compound.
3. The description of the structures in Figures 1 and 6 are not the same.
4. On page 3 they indicate the following "As shown in Figure 6, it can be seen that the conformation of the molecule interacts with the enzyme through hydrophobic interactions with Thr51, Cys57, and Leu334 amino acids residues", could also be present an aromatic amino acid that favors an interaction π- π, will this be possible?.
1. The bibliography must be reviewed carefully, the way the name of some journals is abbreviated is not correct. In some points are not used in others yes.
The manuscript has been well structured, easy to understand, the reported data have been clearly described and discussed.
Author Response
Thank you for your time and considerations about our article. Find bellow the point-to-point answer to the comments.
The authors report the isolation, characterization and biological evaluation as leishmanicide of a new lignan isolated from the leaves of the plant Virola surinamensis (Rol.) Warb native to Central and South America.
The manuscript has been well structured, easy to understand, the reported data have been clearly described and discussed.
Based on the above, my opinion is that the manuscript should be accepted, but first the authors should take into account the following recommendations.
1 Indicate in Table 1, what is the meaning of the letter "a" (400 MHz)a, as well as the letter J in the values reported for the HMBC experiment.
R: Fixed, thank you for your input.
2 If the authors are sure that the secondary metabolite obtained has a negative optical rotation, they should first place the instrument in which it was performed and second place a sign (-) before the name of the characterized compound.
R: Fixed. We added the equipment in the topic 4.2 and we adjust the name of the substance.
3 The description of the structures in Figures 1 and 6 are not the same.
R: We appreciate your comment, but proteins and ligands are flexible molecules that can adopt different conformations. During the docking process, both the ligand and the protein can undergo conformational changes to optimize their interactions. This can lead to slight distortions or adjustments in the ligand's stereochemistry, including the furan ring, to achieve better binding. So instead of naming the Figure 7 (-)-5-demethoxygrandisin B, we changed for the most stable configuration of (-)-5-demethoxygrandisin B. Thank you for your input, we appreciate.
4 On page 3 they indicate the following "As shown in Figure 6, it can be seen that the conformation of the molecule interacts with the enzyme through hydrophobic interactions with Thr51, Cys57, and Leu334 amino acids residues", could also be present an aromatic amino acid that favors an interaction π- π, will this be possible?.
R: In the PoseView program is not possible.
1 The bibliography must be reviewed carefully, the way the name of some journals is abbreviated is not correct. In some points are not used in others yes.
R: Thank you for your input. We would like to notify you that we have diligently examined and rectified the journal name abbreviations in the bibliography section, addressing the concerns you raised.
Reviewer 3 Report (New Reviewer)
review report of the manuscript ID "pharmaceutics-2544704".
The article presents the characterization of a grandisin B derivative isolated from Virola surinamensis. The authors measured the compound antileishmanial activity through practical experiments and theoretical calculations.
Major comments:
1-The HRESIMS data of the compound (calculated and observed need to mention in the manuscript), and carefully check the calculated value.
2-The authors need to provide details of the chemical method used to determine the stereo-structure as well as the spectroscopic data of the chemically modified compound. The authors mention "The stereochemistry in the furan ring was established by analyzing the coupling constants (J) of the H-7 and H-7' hydrogens and by chemically shifting the signals of the methyl hydrogens at C-8 and C-8' [22]", is not proof of the compound stereo-structure.
Minor points:
1-The authors need to adjust the writing style of the chemistry section, including italicizing the delta symbol "δH", "δC" , and the coupling constant symbol "J".
2-In the experimental section: The weights of the extracts is too small regarding the initial wt of the plant sample, why?
3-Is there any other isolated compounds? if not, please explain why. Usually, the isolation experiment continues for purifying several compounds.
Minor English revision is needed
Author Response
Thank you for your time and considerations about our article. Find bellow the point-to-point answer to the comments.
The article presents the characterization of a grandisin B derivative isolated from Virola surinamensis. The authors measured the compound antileishmanial activity through practical experiments and theoretical calculations.
Major comments:
1-The HRESIMS data of the compound (calculated and observed need to mention in the manuscript), and carefully check the calculated value.
R: Dear reviewer, both, calculated mass review and mention were addressed in the main text and supplementary material. Thank you for your insightful feedback.
2-The authors need to provide details of the chemical method used to determine the stereo-structure as well as the spectroscopic data of the chemically modified compound. The authors mention "The stereochemistry in the furan ring was established by analyzing the coupling constants (J) of the H-7 and H-7' hydrogens and by chemically shifting the signals of the methyl hydrogens at C-8 and C-8' [22]", is not proof of the compound stereo-structure.
R: There was a misconception in stating that the stereochemistry was determined by ‘’analyzing the coupling constants (J) of the H-7 and H-7' hydrogens and by chemically shifting the signals of the methyl hydrogens at C-8 and C-8' [22]’’. This does not provide evidence of the compound's stereostructure. We have corrected this statement in the paper, and in the discussion section, we present all the data that support the accurate stereochemistry of the furan ring.
Minor points:
1-The authors need to adjust the writing style of the chemistry section, including italicizing the delta symbol "δH", "δC", and the coupling constant symbol "J".
R: Thank you for the input. We fixed.
2-In the experimental section: The weights of the extracts are too small regarding the initial wt of the plant sample, why?
R: The obtained masses of the hexane, EtOAc, and methanol extracts were 61.7g, 120g, and 244g, respectively. We selected a small sample of EtOAc (6.0g) for our experimental investigations. We have incorporated this information into the experimental section along with a flowchart, as per the request of another reviewer. We appreciate your valuable suggestions.
3-Is there any other isolated compounds? if not, please explain why.
R: Yes, there are at least 15 substances isolated from the ethyl acetate phase, many of them already well established in the literature such as surinamesin and violin. In this paper, our focus was to present the biological activity of just (-)-5-demethoxygrandisin B. Thank you.
Minor English revision is needed
R: Thank you.
Reviewer 4 Report (New Reviewer)
The manuscript presented by Souza Paes et al. addresses the potential of a natural compound of the lignan family (5-demethoxygrandisin B) as an antiparasitic.
The compound is obtained from a natural source and is characterized by spectroscopic techniques and subsequently, in vitro studies are performed. Finally, the authors performed a series of in silico studies to corroborate the potential of the natural compound.
As for the introduction, it is a text that summarizes all the topics of the manuscript, but it is not an introduction that really justifies and highlights the context of the work. This reviewer has several questions that the authors should clarify and should be reflected in the manuscript.
The compound 5-demethoxygrandisin B.
Is this the first time this compound has been isolated or has it been previously described? This topic should be included in the introduction as it is very important to know if the compound is being described for the first time or if it is the first time the genus or species has been described. Are there other structurally related compounds? Have activities been described for this or other related compounds?
As for the target, it does not seem that the information included by the authors in the introduction is sufficient to justify that this is the specific target of the natural compound. Please better contextualize the motivation for your study in that regard.
The paragraph on in silico approaches does not provide any new information nor does it justify the use of the methodologies. This part should either be deleted or modified to justify the use of the methodologies specifically in this study.
Regarding the results, it is considered necessary to include a scheme of how the fractionation process has been carried out from the extract to the final compound, indicating the yields of each extract/solvent.
This reviewer believes that the spectroscopy table should be part of the methodology or supplementary material. The text is adequate and reflects the evidence of structural elucidation. In this text, the letters referring to signals, e.g. s for singlet, should be in italics. Please correct this error throughout the paragraph.
Figure 1 has a mistake in the stereochemistry. The chiral center is position 7 of the furan and not carbon 1 of the trimethoxyphenyl rest. This is a serious error that should be corrected.
Table 2 shows the CC50 and IC50 values in μg/mL and also values in brackets expressed in μM. Is it the corresponding value calculated in that unit? Please clarify in the text what you mean by that value in brackets.
Please indicate the objective of nitrite quantification studies.
In Figure 6, the stereochemistry of the compound is not the same as in Figure 1 where the proposed structure of the isolated natural compound is shown. The authors should propose a new figure with the correct stereochemistry. Therefore, the text concerning this part of the manuscript could be substantially modified.
In view of these evaluations, I recommend that the authors carry out a major revision of the manuscript to improve its quality and be able to consider it for publication in the journal Pharmaceutics
The quality of English is acceptable. A general proofreading could be carried out,
Author Response
Thank you for your time and considerations about our article. Find bellow the point-to-point answer to the comments.
The manuscript presented by Souza Paes et al. addresses the potential of a natural compound of the lignan family (5-demethoxygrandisin B) as an antiparasitic.
The compound is obtained from a natural source and is characterized by spectroscopic techniques and subsequently, in vitro studies are performed. Finally, the authors performed a series of in silico studies to corroborate the potential of the natural compound.
As for the introduction, it is a text that summarizes all the topics of the manuscript, but it is not an introduction that really justifies and highlights the context of the work. This reviewer has several questions that the authors should clarify and should be reflected in the manuscript.
The compound 5-demethoxygrandisin B.
Is this the first time this compound has been isolated or has it been previously described? This topic should be included in the introduction as it is very important to know if the compound is being described for the first time or if it is the first time the genus or species has been described. Are there other structurally related compounds? Have activities been described for this or other related compounds?
R: Indeed, it is the first time this compound is isolated. During its initial submission, we characterized it as an original compound. However, the initial reviewer highlighted the significance of designating the compound as 5-demethoxygrandisin B, since the difference between our compound and 5-demethoxygrandisingrandisin (a substance found in the genus Virola is the stereochemistry of the furan ring). We accepted his correction. But since your insights are extremely relevant, we changed a few things in the introduction as you asked for. I hope you appreciate it.
As for the target, it does not seem that the information included by the authors in the introduction is sufficient to justify that this is the specific target of the natural compound. Please better contextualize the motivation for your study in that regard.
R: Fixed
The paragraph on in silico approaches does not provide any new information nor does it justify the use of the methodologies. This part should either be deleted or modified to justify the use of the methodologies specifically in this study.
R: It was erased. Thank you
Regarding the results, it is considered necessary to include a scheme of how the fractionation process has been carried out from the extract to the final compound, indicating the yields of each extract/solvent.
R: Thank you for your input, a scheme of the obtation of the extracts and the isolation of the substance was made.
This reviewer believes that the spectroscopy table should be part of the methodology or supplementary material. The text is adequate and reflects the evidence of structural elucidation. In this text, the letters referring to signals, e.g. s for singlet, should be in italics. Please correct this error throughout the paragraph.
R: We appreciate your insight. Choosing to retain the table within the body of the article to preserve the substance's structure is undoubtedly a justified decision, since is an unpublished substance and we've been saving our best studys to publish here in pharmaceutics as we did before. This gains even more significance when considering the enhancement of the quality and integrity of the work, a viewpoint reinforced by the consensus of four other reviewers. We value your suggestion. The letters referring to signals is fixed. Thank you very much for your input
Figure 1 has a mistake in the stereochemistry. The chiral center is position 7 of the furan and not carbon 1 of the trimethoxyphenyl rest. This is a serious error that should be corrected.
R: Fixed
Table 2 shows the CC50 and IC50 values in μg/mL and also values in brackets expressed in μM. Is it the corresponding value calculated in that unit? Please clarify in the text what you mean by that value in brackets.
R: Thank you for your inquiry. To ensure clarity and prevent confusion, we have opted to use exclusively the unit µM (micromolar), which has been consistently applied throughout the manuscript.
Please indicate the objective of nitrite quantification studies.
R: The objective of the nitrite quantification studies was to establish that the compound does not act indirectly through macrophage activation. Based on this evidence, we infer that the compound has a direct effect on the parasite, justifying the performance of additional experiments on the promastigote form of the parasite, including Transmission Electron Microscopy and analysis of the Mitochondrial Membrane Potential.
In Figure 6, the stereochemistry of the compound is not the same as in Figure 1 where the proposed structure of the isolated natural compound is shown. The authors should propose a new figure with the correct stereochemistry. Therefore, the text concerning this part of the manuscript could be substantially modified.
R: We appreciate your comment, but proteins and ligands are flexible molecules that can adopt different conformations. During the docking process, both the ligand and the protein can undergo conformational changes to optimize their interactions. This can lead to slight distortions or adjustments in the ligand's stereochemistry, including the furan ring, to achieve better binding. So instead of naming the Figure 6 (-) 5-demethoxygrandisin B, we changed for the most stable configuration of (-) 5-demethoxygrandisin B.
In view of these evaluations, I recommend that the authors carry out a major revision of the manuscript to improve its quality and be able to consider it for publication in the journal Pharmaceutics
R: We did. Thank you for your contributions for our paper.
The quality of English is acceptable. A general proofreading could be carried out.
R: Thank you. We made it
Reviewer 5 Report (New Reviewer)
The article “5-demethoxygrandisin B from Virola surinamensis (Rol.) Warb. Leaves: Evaluation of the Leishmanicidal Activity by In Vitro and In Silico Approaches”. Leishmaniasis is a complex disease caused by Leishmania infection and is transmitted by the bite of an infected female mosquito. The number of drugs used to treat it is still limited, and drug discovery is a valuable approach. Plant-based herbal preparations are often used to treat many diseases. The leaves and seeds of V. surinamensis contain a high content of lignans, well known as a promising class of compounds against Leishmania spp. Among the molecular targets used to find new drugs against Leishmania, trypanothion reductase (TryR) has been identified as unique to the parasite. This article describes the in vitro leishmanicidal activity and in silico interaction between trypanothion reductase (TryR) and 5-demethoxygrandisin B from the leaves of Virola surinamensis (Rol.) Warb. The compound 5-demethoxygrandisin B was isolated from the leaves of V. surinamensis, a plant native to the Brazilian Amazon. The methods used were spectroscopy of HRMS and NMR for Structural Characterization, Peritoneal Macrophage Isolation and Parasite Cultures, Cytotoxicity assay, Antileishmanial Activity Assay and Selectivity Index, Nitrite Quantification, Transmission Electron Microscopy, Determination of Mitochondrial Membrane Potential (MMP)(ΔΨm), Molecular Docking, Binding Free Energy Calculations and others. The results showed that (7R,8S,7'R,8'S)-3,4,5,3',4'-pentamethoxy-7,7'-epoxylignan, namely 5-demethoxygrandisin B isolated from the leaves of V. surinamensis , proved to be selective against parasites that inhibit the promastigote and intracellular amastigote forms of L. amazonica. The antiparasitic effect of this compound was confirmed by ultrastructural changes with a decrease in mitochondrial membrane potential in L. amazonian promastigotes. Docking analysis and molecular dynamics modeling revealed a possible mechanism for TryR inhibition mediated by the 5-demethoxygrandisin B. enzyme.
Overall, the article is a comprehensive, well-described study with good design and discussion. The abstract well reflects the meaning and purpose of the work, the actual results obtained, and the conclusions. Ethical approval obtained. The results are well illustrated in 3 tables and 9 figures. The list of references is represented by 82 relevant sources, mostly of recent years. This study is extremely relevant for practical healthcare and can be recommended for publication without changes.
Author Response
We sincerely appreciate the time and effort invested by the reviewer in reviewing our manuscript.
Round 2
Reviewer 3 Report (New Reviewer)
The authors responded o the comments, however, still some more adjustments are needed:
- The calculated and measured mass was reversed in the supplementary materials.
- Please discuss the compound identification in one section (the results or the discussion).
- Please change δH and δC into italic δ and subscript H or C.
Acceptable
Author Response
The point-by-point response is attached.

Reviewer 4 Report (New Reviewer)
The authors have addressed all the comments of the review and substantially improved the manuscript. I recommend its publication in the present form
Check the final version for minor flaws
Author Response
The authors have addressed all the comments of the review and substantially improved the manuscript. I recommend its publication in the present form.
Thank you for your positive feedback and recommendation. We appreciate your thorough review of our manuscript.
Check the final version for minor flaws.
We have carefully reviewed the final version and addressed any minor flaws. Thank you for your suggestion.
This manuscript is a resubmission of an earlier submission. The following is a list of the peer review reports and author responses from that submission.
Round 1
Reviewer 1 Report
In the present manuscript, Paes et al. investigate the leishmanicidal activity of 5-demethoxygrandisin B isolated from leaves of V. surinamensis. Using in vitro and in silico methods, the authors show the effect of the compound on L. amazonensis pro- and amastigote forms and propose inhibition of trypanothione reductase as the mode of action.
Although the content is certainly of scientific value and the experiments seem to be conducted accurately, I was disappointed by the huge number of mistakes. Each of it might be considered minor but the sheer number is by far too high, especially considering 14 co-authors who claim (Author Contributions) to have reviewed and agreed (i.e., proofread) to the manuscript prior to submission, but obviously did not do so. I felt to be misused by the 14 co-authors having outsourced the proofreading to the reviewer. In addition, I kind of doubt that 14 co-authors have substantially contributed to such a manuscript but are rather listed as a courtesy for whatever reason. The manuscript is in many aspects unprecise and needs to be reworked thoroughly including language check.
Specific Comments
- Please consider again the number of co-authors and if their contribution was substantial and if some might be rather moved to the Acknowledgement.
- A recent review by: Dinesh Kumar Patel: Grandisin and its therapeutic potential and pharmacological activities: Pharmacological Research - Modern Chinese Medicine 5 (2022) 100176, is already mentioning these kind of structures and their potential towards Trypanosoma and Leishmania. Please carefully consider this manuscript and use it as a reference to validate your research also for already discussed results mentioned elsewhere. Also, please do not use at all the word “new” in front of chemical compound as done on p. 1, abstract and throughout the whole text.
- p. 8, 9 figure 5: Scatter plot is mentioned but not discussed/interpreted at all. What does it tell us? What are the colors in the plot referring to? In general, paragraph 2.5 and figure 5 subtext are not very clear. Please rework. 60° treated parasites are rather a positive than a negative control. What are the colors and the horizontal lines in the figure referring to? What is the purpose of the TMRE staining?
- Throughout the text: different abbreviations used for Trypanothione Reductase (TR and TryR), please correct
- p. 1, abstract: V. surinamensis (Rol.) Warb. or only “V. surinamensis”, also at end of introduction p. 2
- p. 1, abstract: “Cytotoxic and nitrite production…”: missing word? à “analyzed” instead of “carried out”
- p. 1, abstract: give IC50 values in molar range
- p. 1, abstract: “compacting patterns”, “membranes”, “autophagosames”
- p. 1-2, abstract: “vesicular electron-dense structures with outer bilayer membrane, structure-like-autophagosome (???), and enlargement of flagellar pocket containing filaments”: sentence not clear, please rephrase
- p. 2, abstract: binding free energy values cannot show potential activity on the parasite
- p.2, CADD methods: it is certainly true that these approaches are very valuable methods in the DD process, however, they are not “most outstanding” and also not “promising alternatives to traditional experimental methods”, but rather supportive methods
- p. 2, [15] has been withdrawn à not possible to cite
- p. 3, introduction: what is the rational for choosing trypanothione reductase?
- p. 5, l. 1: “activity was analyzed” instead of “performed”
- p. 5: The conclusion is drawn that promastigotes are more sensitive than the intracellular amastigotes. This could also be due to lower bioavailability of the compound in the cell
- p.5: “26.6 and 7.4 times less toxic”
- p.5, header table 2: “Cytotoxicity”
- p. 6, table subtext: “macrophages”
- p. 6, “in the supernatant of BALB/c peritoneal macrophages”
- p. 7, figure 3: “5-demethoxygrandisin B” instead of “7,7´-expoxylignan”
- p. 7, figure 3 subtext: please mention also LPS
- p. 7, last paragraph: what is meant by “enlargement of flagellar pocket containing filaments”?
- p. 8, first line last paragraph: “In figure 5”
- p. 9, paragraph 2.6 first sentence, same as introduction: What is the rational for choosing TryR? The compound could, in principle, have antileishmanial activity whilst not acting on TryR. The more elaborate explanation is found in the discussion, of course, for the reader it would be convenient to have a brief explanation already in the results section
- p. 9, paragraph 2.6: L. amazonenis in vitro experiments are compared to L. infantum in silico experiments. Although it is expected that there is no fundamental (if at all) difference in the active site of the enzymes of both species, this should be secured by a sequence alignment
- p. 10, figure 6: Please add the length of the H-Bonds
- p. 10: “2.7 Molecular Dynamics”
- p. 10, paragraph 2.7: what is “RT-ligand complex”? Is this referring to trypanothione reductase?
- p. 12, “2.8 Binding Free Energy Calculations”
in first line of paragraph: “Molecular Mechanics Generalized…”
further down: “by comparing the binding free energy values of these two methods it is possible…”
please check abbreviations in brackets, they do not match with table 3
- p. 12, “2.9 Per-residue Energy Decomposition”
- p. 13, figure 9: please explain color bar
- p. 14, last paragraph: what is meant by “enlargement of flagellar pocket containing filaments”?
- p. 15, second paragraph: “with compounds like lignans”
- p. 15, second paragraph: “by competitively inhibiting”
- p. 15, fifth paragraph: “N-terminal portion (48-87)”, where do these numbers come from? On p. 11 figure 8 it says “Ser76-Asn91” for the region of highest flexibility.
- p. 19, second paragraph: what is “RT structure”? Is this referring to trypanothione reductase?
- p. 19, third paragraph: “Using the leap module available…”: Why did the solvation with TIP3P water molecules add charges, that needed to be neutralized? It probably did not, and the charges resulted from the ligand? Please rephrase.
- p. 19, last paragraph: “4.11.1 Binding Free Energy Calculations”
- p. 19, conclusions: “V. surinamensis (Rol.) Warb.”
language check needed
Reviewer 2 Report
The authors purified and characterized an original natural compound with moderate leishmanicidal activity and selectivity. The performance of experiments is impeccable. However, the use of just a single compound confers a limited value to this study. The section 2.3 and Fig. 3 seem to be insufficiently related to the rest of the work. More importantly, the in silico data are insufficient for the suggestion about the possible inhibition of trypanothione reductase and its role. This should be supported by the experiments using either purified enzyme either the extracts of parasites.